# Revision Total Hip Arthroplasty Utilizing an Acetabular Reinforcement Ring with a Metal Augment: A Minimum Eight-Year Follow-Up Study

**DOI:** 10.3390/medicina59061036

**Published:** 2023-05-27

**Authors:** Han Soul Kim, Ji Wan Kim, Jae Suk Chang, Chul-Ho Kim

**Affiliations:** 1Department of Orthopedic Surgery, Gachon University Gil Medical Center, Incheon 21565, Republic of Korea; hankhan118@gmail.com; 2Department of Orthopedic Surgery, Asan Medical Center, University of Ulsan College of Medicine, Seoul 05505, Republic of Korea; bakpaker@hanmail.net; 3Department of Orthopedic Surgery, National Police Hospital, Seoul 05715, Republic of Korea; jschang3525@gmail.com

**Keywords:** acetabular reinforcement ring, metal augment, hip arthroplasty, revision

## Abstract

*Background and Objectives:* An acetabular reinforcement ring (ARR) with a structural allograft is conventionally used to treat large acetabular bone defects or discontinuity during revision hip arthroplasty. However, ARR is prone to failure due to bone resorption and lack of incorporation. Here, we investigated the surgical outcomes of the patients who underwent revision total hip arthroplasty (THA) using ARR combined with a metal augment (MA). *Materials and Methods:* We retrospectively reviewed data from 10 consecutive patients who had a minimum 8-year follow-up after revision hip arthroplasty using ARR with MA in Paprosky type III acetabular defect. We collected patient demographics, surgical details, clinical scores (including Harris Hip Score (HHS)), postoperative complications, and 8-year survival rates. *Results:* Six male and four female patients were included. The mean age was 64.3 years, and the mean follow-up duration was 104.3 months (96.0–112.0 months). Trauma-related diagnosis was the most common reason for index surgery. Three patients underwent all component revision, and seven underwent cup revision. Six were confirmed as Paprosky type IIIA and four as type IIIB. The mean HHS at the final follow-up was 81.5 (72–91). One patient was diagnosed with prosthetic joint infection at the 3-month follow-up; therefore, the minimum 8-year survival rate with our technique was 90.0% (95% confidence interval, 90.3–118.5%). *Conclusions:* The satisfactory mid- to long-term results of revision THA suggest that ARR combined with tantalum MA is a viable revision option for treating severe acetabular defects with pelvic discontinuity.

## 1. Introduction

Total hip arthroplasty (THA) is one of the most successful procedures in orthopaedics, and with the rising incidence of THA worldwide, revision surgeries are becoming more common [1,2]. Indeed, the volume of revision THA is expected to grow along with the economic burden of this procedure [2,3]. Although acetabular component revision is a relatively rare cause of revision, one study suggests that cup loosening is the most common mode of failure in elderly patients, and the authors believe that the numbers will only grow with increased life expectancy [4]. Although significant developments, such as the implementation of robotic surgery and three-dimensional printing techniques, have been made in revision hip arthroplasty [5,6,7], revision in severe acetabular defects can still be challenging [1,8].

In conditions of severe bone deficiency around the acetabulum, complex acetabular reconstruction procedures are required to restore the acetabulum and to provide sufficient bony support for the implant [9]. Various surgical options for the revision of severe acetabular bony defect include the high hip centre technique, the bilobed cup or the jumbo cup, structural allograft or impaction bone graft, and a cage or a ring [10,11,12,13,14]. The acetabular reinforcement ring (ARR) has been widely used to resolve severe bone defects and pelvic discontinuity in revision THA [15,16]. ARR protects bone grafts and bridges the gap between the superior and inferior portions of the disconnected pelvis [15,16,17,18]. Studies utilising ARRs and structural bone grafts have reported favourable results in short to midterm follow-up periods [18,19]; however, the incidence of reoperation and radiographic loosening are known to be higher than for the other options, such as porous tantalum metal augment (MA) [14]. ARR is commonly used in surgery together with structural allograft [15,20,21,22,23]. In this case, allogenic bone fills the bone defect, forming a bone bridge, and aiding in the restoration of pelvic discontinuity and mechanical stability. However, the bone grafts around ARR often result in failure due to bone resorption or lack of incorporation [21,24]. Therefore, the use of ARR alone carries the risk of implant breakage and early loosening due to the lack of biologic incorporation.

Recently, the introduction of porous tantalum MA has been a paradigm shift in revision THA. While serving as a void filler in multiple different shapes, it has excellent innate properties that can promote both primary biologic fixation and supplementary fixation for adjunctive implants [13]. Porous tantalum MA is now being widely used with promising midterm results [25,26,27,28,29,30,31], which are fortified by a recent systematic and meta-analysis review that investigated the safety and effectiveness of MA [32]. However, despite these advantages, there are cases where an acetabular defect combined with pelvic discontinuity is so large that using conventional MAs or a jumbo cup alone is a challenge, and ARR is necessary in order to bridge the pelvic bone [29]. To address these issues, few studies have utilized a porous tantalum metal cup with a cage and reported moderate midterm results [25,28]. However, the cup–cage construct still needs allograft before placing the tantalum metal cup, which may lead to early failure due to the aforementioned disadvantages of allografts. It is only recently that some studies have reported short- to midterm outcomes after the use of a cage with MA instead of with a bulk allograft [33,34,35].

Thus, we investigated the following question: Can a combination of ARR and MA demonstrate satisfactory longevity with stable fixation? Under the hypothesis that the use of an ARR with MA will provide stable fixation and a satisfactory outcome, we investigated the surgical outcomes of a minimum 8-year long-term follow-up in patients who underwent revision THA using ARR and MA. To our knowledge, this study reports the longest midterm results of revision using ARR with MA.

## 2. Materials and Methods

This study was approved by the Institutional Review Board of our institution, and a waiver was received for the need to provide written informed consent of the participants. Data collection was performed in accordance with the relevant guidelines and regulations of the committee.

### 2.1. Patient Selection and Data Collection

We performed a retrospective review of consecutive patients who underwent revision THA using MA combined with ARR in our institution between January 2014 and March 2015. We chose the cup–cage system when the size of the acetabular defects were extensive for the jumbo cup (males over 64-mm, females over 60-mm) [36] to cover. We assessed the acetabular defect using preoperative 3D-CT in all revision THA cases, and we prepared ARR with MA for the cases that showed the acetabular defect with more than a 2 cm acetabular component migration, which means Paproski classification Type III or IV.

The inclusion criteria were: (1) patients who underwent revision THA using ARR with MA due to severe acetabular defects classified as Paprosky type III [37], and (2) a minimum follow-up of 8 years. Ten patients were included in the study (Figure 1).

Demographic data, including the sex, age, body mass index (BMI), and bone mineral density (BMD) of the patients at the time of final revision surgery; the reason for indexing and following revision surgeries; and the data of the follow-up duration were collected. The investigated perioperative details were as follows: affected side, components detail used in surgery, total number of surgeries prior to the final revision THA, intraoperative Paprosky classification of the acetabulum at the final revision stage, and type of implant and the surgical techniques implemented. For postoperative details, we investigated the final Harris Hip Score (HHS) in order to evaluate clinical outcome and the presence of postoperative complications as surgical outcomes.

### 2.2. Surgical Technique

All of the procedures were performed by a single senior surgeon via a modified Hardinge approach or posterior approach, with the patient in a lateral decubitus position. Sliding osteotomy of the greater trochanter or extended trochanteric osteotomy was performed when appropriate for better acetabular exposure. The acetabulum was evaluated preoperatively using three-dimensional computed tomographic (3D CT) images and intraoperatively using Paprosky classification [38] after preparing the acetabular defect by reaming and curettage. A trabecular metal augment (Zimmer-Biomet, Warsaw, IN, USA) and a Burch-Schneider (BS) reinforcement cage (Zimmer-Biomet, Warsaw, IN, USA) were applied in all cases.

The appropriate size and shape of the MA was first selected. The MA was then fixed with cancellous screws and filled with morselised bone graft. The ilium and ischium were exposed, and the two flanges of the BS cage were pre-shaped according to individual pelvic contour. The cage was firmly fixed superiorly onto the ilium using multiple screws, and inferiorly into the ischial slot. Additional stability was promoted via fixing the cage onto the MA with bone cement. We inserted the maximum number of screws possible into locations with remaining bone stock. Finally, a low-profile polyethylene acetabular liner was cemented onto the cage with an appropriate degree of anteversion and inclination. Careful attention was paid to avoid neurovascular injury during the entire procedure. The details of the surgical procedure are schematised in Figure 2.

### 2.3. Postoperative Management

Progressive tolerable weight-bearing using walking aids was permitted from day one depending on the patient’s individual capability. Full weight-bearing without assistance was permitted from the postoperative period of 6–12 weeks as radiologic stability was confirmed at the outpatient session. The patients were reviewed clinically and radiologically at 6, 12, and 24 weeks, at 1 year, and then every year thereafter. Radiologic stability and osteointegration of the ARR and MA construct were evaluated at each outpatient session by assessing (1) radiolucent lines of >2 mm around the acetabular component in three zones, as described by DeLee and Charnley [39], (2) screw breakage, (3) acetabular cup migration greater than 2 mm, and (4) inclination angle change greater than four degrees, as described by Lim et al. [40].

### 2.4. Statistics

The Statistical Package for Social Sciences software (IBM SPSS Statistics, version 18.0, for Windows, Chicago, IL, USA) was used for data analysis. Descriptive statistical results were recorded to describe all outcomes.

## 3. Results

### 3.1. Patient Demographics

The study enrolled six male and four female patients. The mean age at revision surgery was 64.3 years (44–80 years) and the mean BMI was 25.6 kg/m^2^ (19.5–34.1 kg/m^2^). Four patients were diagnosed with osteoporosis. A total of 7 out of the 10 patients underwent initial arthroplasty surgery due to trauma-related problems, such as traumatic osteoarthritis following acetabular fracture, traumatic osteonecrosis of femoral head, or acute femoral neck fracture. The most common reason for revision arthroplasty using ARR with MA was component loosening of prior hip arthroplasty: 9 out of the 10 patients were diagnosed with cup loosening, of which 3 patients confirmed all component loosening and stem loosening. The number of revision surgeries from initial surgery to final revision using ARR with MA was variable from the first to the fourth. The mean follow-up duration of the 10 enrolled patients was 104.3 months (96.0–112.0 months). The details of the patient demographics are described in Table 1 below.

### 3.2. Surgical Details and Postoperative Results

The surgical details and outcomes are described in Table 2. In Cases 1 and 2, stem revision was performed due to stem loosening. In Case 5, although the femoral component was well fixed, an all-component revision was necessary to replace the initial monobloc stem. Intraoperative assessment of Paprosky classification of the acetabulum revealed six type IIIA and four type IIIB cases. The size of the BS cage ranged from 50–62 mm. A mesh was used to cover the medial wall defect in two cases (Cases 1 and 10). Sliding osteotomy of the greater trochanter was performed in five cases, and extended trochanteric osteotomy was performed in one case in order to facilitate acetabular exposure (Case 5). At the final follow-up, the mean HHS was 81.5 points (range, 72–91, SD 6.6). Additional surgical details are described in Table 2. A favourable treatment outcome using ARR combined with MA is shown in Figure 3.

During the follow-up period, two cases of postoperative complications were observed. Case 3 was diagnosed with prosthetic joint infection at postoperative 3-month follow-up; therefore, the patient underwent implant removal and a girdlestone procedure (Figure 4). Case 3 was recorded as a treatment failure. Case 4 was readmitted due to hip dislocation 4 weeks postoperatively, but after capsular reinforcement surgery was performed and a hip abductor brace was applied, no recurrent dislocation occurred. Thus, the 8-year survival rate of revision arthroplasty using ARR combined with MA was 90.0% (95% CI: 90.3–118.5).

## 4. Discussion

Acetabular reconstruction in a rare case of large bone defect with pelvic discontinuity is challenging. Although numerous options exist for restoring this defect, a single method has not been proven superior to others. To bridge a large gap between the pelvis and fill a large void in the defect, a construct needs to ensure rigid fixation and bony ingrowth. We hypothesized that ARR with MA will provide stable fixation and a satisfactory outcome, and we investigated the surgical outcomes of a minimum 8-year long term follow-up in patients who underwent revision THA using ARR and MA. This study, including 10 revision THA cases using ARR combined with MA, revealed favourable long-term results at minimum 8-year follow-up at a survival rate of >90.0%, with acceptable clinical outcomes as 81.5 points of the mean HHS.

ARR is indicated when the acetabular component has <50% contact with the host bone during revision THA [15,18]. Traditionally, bone grafting techniques have been used concomitantly for filling in the remaining acetabular bone defects [15,17,18,20,22,23,41]. Reports on using ARR with bone graft have been favourable: Hsu et al. and Regis et al. reported survival rates of 76% and 80% and a final HHS of 67 and 75.6 points, respectively, in their 3–10-year-follow-up study, for example [20,22]. Although structural allografts restore bone stock and hip centre and thus yield acceptable results, they are technically challenging and prone to failure due to bone resorption and a lack of incorporation. Several studies have reported the consecutive failure of ARR with the lysis of the bone graft [42], and increased the risk of transmitting infection or septic complications [21,24]. For this reason, a senior surgeon at our institution searched for options to replace allografts. Due to concerns regarding bone resorption and ARR failure, we had been avoiding the use of ARR alone before the study period. With the availability of MA in our institution since January 2014, we have been using a combination of ARR with MA in all cases that required the use of ARR.

Here, we found that MA utilization with ARR yielded a survival rate of >90.0% with favourable clinical outcome scores of >8 years of follow-up. When survival is defined as another revision performed due to loosening or mechanical failure, as in other studies, our survival rate increases to 100%. Porous metal augments offer several advantages over structural allografts, such as easier surgical technique and no risk of disease transmission [27,43]. In addition, porous metal augments are reliable for bone ingrowth, and they will not be resorbed [43,44]. Consequently, MA adds stability onto the cage, allowing for better survival.

According to systemic and meta-regression analyses, loosening was reported in 11 out of 37 cases from five studies that used ARR for pelvic discontinuity [29]. Our findings are promising in that aseptic loosening was not observed in the 10 pelvic discontinuity revision cases. Lachiewicz et al. reported a survival rate of 96% using tantalum acetabular components in 26 hips with Paprosky type III acetabular defects, which was during a mean follow-up of 3.3 years [26]. Similarly, Davies et al. reported no loosening or mechanical failure in 36 hips with Paprosky type III during a mean follow-up of 4.2 years [45]. Del Gaizo et al. reported one failure in 37 hips with Paprosky type III during a mean follow-up of 3.7 years [44]. However, these studies included only four pelvic discontinuity cases with mean follow-ups of <5 years. Hence, a major strength of our study is that all of the 10 cases included patients with pelvic discontinuity and large acetabular defects with a mean follow-up of >8 years.

Three other groups have recently reported the results of implementing ARR combined with MA [33,34,35]. Makinen et al. reported acceptable results in 22 hips, among which only 7 were cases with pelvic discontinuity, with a mean follow-up of 39 months (27 to 58) [35]. All of the three failed cases were due to asceptic loosening, and all had pelvic discontinuity. The overall 36-month Kaplan–Maier (KM) survivorship was 90.9% (95% CI 73.9 to 107.9), and the 55-month survivorship was 75.7% (95% CI 38.1 to 113.3) [35]. Baeker et al. showed 90% survivorship (two revisions due to cage inferior flange loosening) of the construct in 20 patients with a mean follow-up of 2.8 years [33]. In Garceau et al., 41 patients with a mean follow-up of 6.4 years (range 2.8–11.0) showed 10 year KM survivorship of 87.4% (95% CI 75.3–99.6) [34]. Survivorship was 45.0% for 11 patients with discontinuity at 10 years. Overall, four asceptic cage loosenings were recorded.

Our clinical scores, with a mean HHS of 81.5 points, were satisfactory. Studies using ARR have reported a mean HHS between 76.6 and 86 points [25,26,44,45]. Makinen et al. reported a mean postoperative Oxford Hip Score of 28.7 (13 to 38) at the last follow-up [35]. Baeker et al. reported that the mean HHS improved from 35.3 (range, 13.8–67) to 77.3 (range 38–99) [33].

The complications seen in this study included one case of infection and one case of dislocation. Dislocation is a commonly reported complication after performing revision surgeries using ARR. In fact, the dislocation rates in four of the studies discussed varied from 5% to 9% [25,26,44,45]. This is attributed to weak abductor muscle power, possibly caused by multiple operations damaging the muscle fibres, extensive soft tissue injury leading to a superior gluteal nerve during revision, or greater trochanter non-union [26,46]. Studies have suggested the use of a larger femoral head and abduction brace orthosis postoperatively in order to minimise dislocation [20,26]. Another study has highlighted the need for constrained-type cup liners in order to prevent dislocation, although this is not routinely recommended [18]. Gross et al. reported a decreased dislocation rate using modified trochanteric osteotomy, in which the posterior aspect of the greater trochanter is saved, leaving the external rotators and the posterior capsule intact [17]. In our case, one patient with dislocation was managed with capsular reinforcement and with application of a postoperative hip abductor brace. The patient experienced no further adverse events. Patients in the study by Makinen et al. included one sciatic nerve palsy, one hematogenous infection, one greater trochanteric fracture, one dislocation, and one heterotopic ossification [35]. Baeker et al. reported one intraoperative femoral artery injury, one peroneal nerve palsy that was resolved at the last follow-up, two deep infections, one femoral component loosening, one deep vein thrombosis, and two dislocations [33]. Complications in Garceau et al. included seven deep infections, four dislocations, four sciatic nerve injuries, one vascular injury, two heterotopic ossification, and five cage fractures [34]. Overall, our findings are in line with other studies that utilized ARR with MA in terms of the survivorship, the complication rate, and the clinical results.

This study has the limitations of using only a small sample size as well as potential bias due to its retrospective nature. Ideally, a prospective comparative study with more cases and a longer follow-up period would strengthen the results. However, due to the rarity of large acetabular defect cases and the complexity of THA revisions, conducting a large-scale prospective study in a single centre appears to be quite difficult. Nevertheless, considering the sparsity of outcome data for revision THA with large acetabular defects, the results of our study on 10 Paprosky type III cases using ARR with MA have great value. Additionally, to our knowledge, this study reports the longest midterm results of revision using ARR with MA. We believe a future multi-centre study with a longer period of follow-up is required to further validate both the outcomes and the complications of revision techniques using ARR with MA.

## 5. Conclusions

The satisfactory midterm and long-term results of revision THA suggest that ARR combined with tantalum MA is a viable revision option for treating severe acetabular defects with pelvic discontinuity.

## Figures and Tables

**Figure 1 medicina-59-01036-f001:**
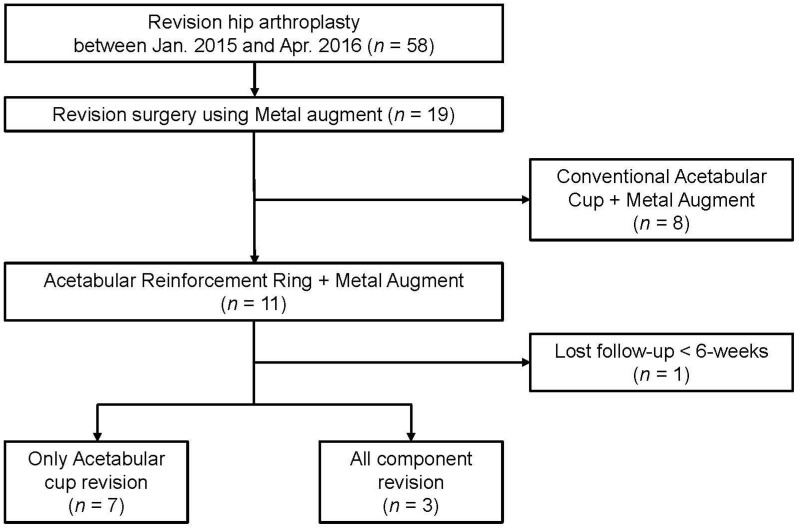
Flow chart of retrospective patient selection for the study.

**Figure 2 medicina-59-01036-f002:**
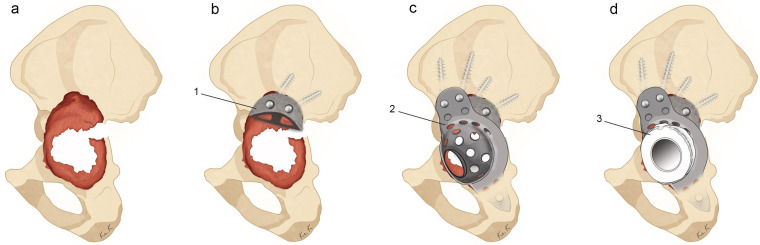
Illustration of revision hip arthroplasty with acetabular reinforcement ring (ARR) combined with a metal augment (MA). Paprosky type IIIB acetabular defects (**a**), MA fixation and bone grafting (**b**), cage fixation, (**c**) and cementing polyethylene liner (**d**). (1: MA, 2: ARR, 3: polyethylene liner).

**Figure 3 medicina-59-01036-f003:**
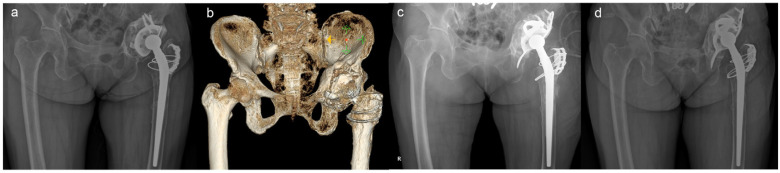
A 71-year-old female patient who underwent acetabular revision with acetabular reinforcement ring and metal augment (case 8). Paprosky type IIIA acetabular aseptic loosening is shown in a simple hip X-ray (**a**) and 3D CT image (**b**). Immediate postoperative X-ray shows stable implant fixation (**c**). Bony consolidation is seen in the 8-year follow-up X-ray (**d**).

**Figure 4 medicina-59-01036-f004:**
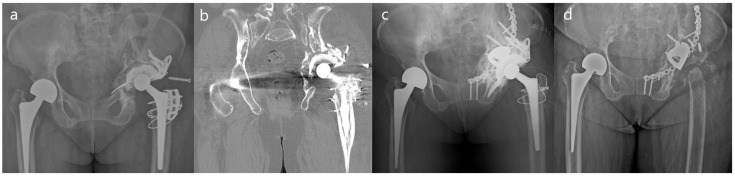
A 69-year-old female patient who underwent acetabular revision with a acetabular reinforcement ring and metal augment (Case 3). Paprosky type IIIB acetabular aseptic loosening is shown in a simple hip X-ray (**a**) and CT image (**b**). Immediate postoperative X-ray shows stable implant fixation (**c**). Implant removal and girdlestone procedure were performed due to prosthetic joint infection at 3-month follow-up (**d**).

**Table 1 medicina-59-01036-t001:** Patient demographics.

Case	Sex/Age	BMI * (kg/m^2^)	BMD * (T-Score)	Reason for Index Surgery	Reason for Final Revision Surgery	No. of Revision Surgery	Follow-Up Duration (Months)
1	F/44	24.1	−1.1 ^†^	Traumatic OA	Cup loosening	2	112.0
2	M/71	21.6	−3.3	Femoral Neck Fx	All component loosening	3	112.0
3	F/69	34.1	−2.1	Traumatic OA	Cup loosening	2	106.7
4	M/65	28.1	−2.6	Femoral Neck Fx	All component loosening + ceramic liner Fx	1	96.0
5	M/58	26.3	−1.0	Traumatic ONFH	Acetabular destruction after monobloc arthroplasty	1	112.0
6	M/50	29.0	0.9	Idiopathic ONFH	Cup loosening	2	107.9
7	M/75	21.7	−1.9	Traumatic ONFH	All component loosening + ceramic liner Fx	3	100.0
8	F/71	26.2	−3.1	Traumatic OA	Cup loosening	4	101.1
9	M/60	24.9	−0.7	Septic Hip Sequalae	Cup loosening	1	99.6
10	F/80	19.5	−2.8	LCP Sequalae	Cup loosening	3	96.0

Abbreviation: F, female; M, male; OA, osteoarthritis; Fx, fracture; ONFH, osteonecrosis of femoral head; LCP, Legg Calve Perthes; PJI, prosthetic joint infection; * data was collected at admission period for final revision surgery; ^†^ Case 1 patients measured BMD by Z-score.

**Table 2 medicina-59-01036-t002:** Surgical details and postoperative results at final revision surgery using acetabular reinforcement ring combined with a metal augment.

Case	Side	Operation	Paprosky	Implants	Additional Procedure	Final HHS	Postop Cx
1	Lt	Total ^a^	3A	B-S + TM + Mesh	GTO + Grip plate	85	-
2	Lt	Total	3A	B-S + TM	Trochanter wiring	91	-
3	Lt	Cup ^b^	3B	B-S + TM	-	73	PJI
4	Lt	Cup	3B	B-S + TM	-	72	Acute D/L
5	Lt	Total	3A	B-S + TM	ETO + Wiring	86	-
6	Lt	Cup	3A	B-S + TM	GTO + Grip plate	90	-
7	Lt	Cup	3A	B-S + TM	-	81	-
8	Lt	Cup	3A	B-S + TM	GTO + Grip plate	79	-
9	Lt	Cup	3B	B-S + TM	GTO + Grip plate	82	-
10	Rt	Cup	3B	B-S + TM + Mesh	GTO	76	-

Abbreviation: Lt, left; Rt, right; ^a^ total revision; ^b^ cup revision; B-S, Burch–Schneider cage; TM, tantalum metal block; GTO, great trochanteric osteotomy; ETO, extended trochanteric osteotomy; PJI, prosthetic joint infection; D/L, dislocation; HHS, Harris Hip Score; Postop Cx, postoperative complication.

## Data Availability

The datasets generated and analyzed during the current study are not publicly available since they contain potentially identificatory information for each patient; but they are available from the corresponding author on reasonable request.

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
