# Peer review of "Revision Total Hip Arthroplasty Utilizing an Acetabular Reinforcement Ring with a Metal Augment: A Minimum Eight-Year Follow-Up Study"

_medicina, 2023, doi:10.3390/medicina59061036_

Round 1

Reviewer 1 Report

Thank you for the great opportunity to review this article. There are several questions that needs to be detailed. 

I understand this study showed that good clinical outcome and less complication during mid to long term follow-up periods. However, I cannot understand the advantage of the acetabular reinforcement ring with metal augmentation. The authors should explain the reason or mechanism for good result of this technique. 

In figure 1, if the ARR was used, was the MA always used?

Are there any cases where only ARR was used or bone graft was used together?

The indication for ARR with MA should be described in more detail.

Although several studies have reported failure or complication of ARR with bone graft, many other studies showed good result.

What is the cause of the failure or complication of ARR? And what is the advantage your technique compare to the ARR with bone graft.

The authors should describe this.

Since the sample size is small, Kaplan Meier curve seems meaningless.

Minor editing of English language required.

Author Response

  1. I understand this study showed that good clinical outcome and less complication during mid to long term follow-up periods. However, I cannot understand the advantage of the acetabular reinforcement ring with metal augmentation. The authors should explain the reason or mechanism for good result of this technique.

Author’s response:

We appreciate your taking the time to review our paper and suggest constructive advices. When the ARR introduced, it usually recommended to use for the conditions of severe bone deficiency around the acetabulum, therefore in many cases it used combined with homologous bone graft.

In similar, the MA originally invented for combination conventional size cup, to avoid complications following to use large acetabular cup (so called, jumbo cup) or ARR.

Herein, we used ARR combined MA, and showed satisfactory results, not the ARR combined homologous bone graft which often result in failure due to bone resorption or lack of incorporation.

We have added the more details about this issue.

Text changes:

ARR is commonly used in surgery together with allogenic bone grafts (e.g., auto- or allografts). In this case, allogenic bone fills the bone defect, aiding in the restoration of mechanical stability… (Page 2, Lines 51-57)

  1. In figure 1, if the ARR was used, was the MA always used?

Author’s response:

We thank you for providing us with the opportunity to comment on this matter. In our institution, due to concerns regarding bone resorption and ARR failure, we had been avoiding the sole use of ARR before the study period. However, since January 2014, the MA was available to use in our institution, we have been using a combination of ARR with MA in all cases involving the use of ARR. Overall study period, we applied MA in all cases which used ARR.

We have added the sentences in the Discussion section to reflect your kind comments.

Text changes:

For this reason, senior surgeon at our institution searched for options to replace allografts. Also, due to concerns regarding bone resorption and ARR failure, we had been avoiding the use of ARR alone before the study period. With the availability of MA in our institution since January 2014, we have been using a combination of ARR with MA in all cases that required the use of ARR. (Page 9, Lines 227-231)

  1. Are there any cases where only ARR was used or bone graft was used together?

Author’s response:

Thank you for the careful attention to detail. As we have response the query No.2, we have no cases of sole ARR without MA during study period.

We did not compare the cases which ARR used without MA in the early period, because there was no consistency in patient selection (there is several years of time-gap exist, between sole ARR usage and ARR + MA technique) and the surgeons were not uniform, which could introduce significant bias. Therefore, please excuse us we have only showed the result of ARR + MA cases in the current study.

Text changes:

None

  1. The indication for ARR with MA should be described in more detail.

Author’s response:

Thank you for your exceptionally valuable comments. As you suggested, we have added the more details about the indication for the ARR with MA in Materials and Methods section.

Text changes:

We chose the cup-cage system when the size of acetabular defects were extensive for jumbo cup (males over 64-mm, females over 60-mm)[36] to cover. We assessed the acetabular defect using preoperative 3D-CT in all revision THA cases, and prepared ARR with MA for the cases which show the acetabular defect more than 2-cm acetabular component migration which means Paproski classification Type III or IV. (Page 2, Lines 87-91)

  1. Although several studies have reported failure or complication of ARR with bone graft, many other studies showed good result.

Author’s response:

We totally agreed your comments. Therefore, we included that information in the main text.

Text changes:

Studies utilising ARRs and structural bone grafts have reported favourable results in short-to-mid-term follow-up periods;[18, 19] however, indicence of reoperation and raiograophic loosening are known to be higher than other options such as porous tantalum metal augment (MA).[14] (Page 2, Lines 48-51)

  1. What is the cause of the failure or complication of ARR? And what is the advantage your technique compare to the ARR with bone graft.

Author’s response:

We thank you for providing us with the opportunity to comment on this matter. Following previous study, the most common reason for ARR failure is mechanical failure due to superior migration following lack of host bone support. And, this could be came from bone resorption and lack of incorporation (Udomkiat P, Dorr LD, Won YY, Longjohn D, Wan Z. Technical factors for success with metal ring acetabular reconstruction. J Arthroplasty. 2001 Dec;16(8):961-9. doi: 10.1054/arth.2001.27669. PMID: 11740749).

Theoretically, MA possesses stronger mechanical properties compared to allogenic bone graft and does not raise concerns about bone resorption or lack of incorporation. Consequently, we believed that MA has advantages, especially during the long-term follow-up period, when compared to ARR with allogenic bone graft. Indeed, several studies have reported favorable results when combining MA with allogenic bone graft. However, there is a lack of studies that have reported long-term follow-up results.

Following your comments, we have added the discussion about this issue more clearly, in Discussion section.

Text changes:

Porous metal augments offer several advantages over structural allografts such as easier surgical technique and no risk of disease transmission.[27, 43] In addition, porous metal augments are reliable for bone ingrowth and will not be resorbed.[43, 44] Consequently, MA adds stability onto the cage allowing for better survival. (page 8, line 235-239)

Recently, the introduction of porous tantalum MA has ben a paradigm shift in revision THA. While serving as a void filler in multiple different shapes, it has an excellent innate properties that can promote both primary biologic fixation and supplementary fixation for adjunctive implants.[13] (page 2, line 58-61)

However, the bone grafts around ARR often result in failure due to bone resorption or lack of incorporation.[21, 24] Therefore, use of ARR alone has the risk of implant breakage and early loosening due to the lack of biologic incorporation. (page 2, line 54-57)

  1. Since the sample size is small, Kaplan Meier curve seems meaningless.

Author’s response:

Thank you for your valuable comment. We agreed your comment, therefore we have removed the K-M curve.

Text changes:

REMOVED figure 4.

  1. Minor editing of English language required.

Author’s response:

Thank you for the careful attention to detail. Even though we initially underwent English editing but applied overall modifications to enhance English language, we checked the entire manuscript again.

Text changes:

Following your suggestion, the entire manuscript underwent revision. Given its length, we have not copied the text into the response letter.

Reviewer 2 Report

why was a harding approach used rather than posterior approach

Does a GT osteotomy or ETO have more morbidity? Posterior approach would have prevented this problem.

Acetabular Impaction bone grafting when done properly has good outcomes. All methods of treating these difficult revisions will have problems.

The method shown or described by the authors is not novel. This technique is performed widely. The results are good.

Overall well written and interesting. It is a good read for orthopaedic hip surgeons. 

Author Response

  1. Why was a Harding approach used rather than posterior approach? Does a GT osteotomy or ETO have more morbidity? Posterior approach would have prevented this problem.

Author’s response:

We appreciate for pointing out this oversight. Indeed, we have used Harding approach as our first option due to surgeon’s preference, but we also used posterior approach with GT osteotomy or ETO (shown in Table 2). We have revised the sentences appropriately.

Text changes:

All procedures were performed by a single senior surgeon, via modified Hardinge approach or posterior approach,… (Page 3, Lines 107-108)

  1. Acetabular Impaction bone grafting when done properly has good outcomes. All methods of treating these difficult revisions will have problems.

Author’s response:

We thank you for providing us with the opportunity to comment on this matter. We agreed your comment that all methods of treating these difficult revisions will have problems.

However, following previous study, the most common reason for ARR failure is mechanical failure due to superior migration following lack of host bone support. And, this could be came from bone resorption and lack of incorporation (Udomkiat P, Dorr LD, Won YY, Longjohn D, Wan Z. Technical factors for success with metal ring acetabular reconstruction. J Arthroplasty. 2001 Dec;16(8):961-9. doi: 10.1054/arth.2001.27669. PMID: 11740749).

Theoretically, MA possesses stronger mechanical properties compared to allogenic bone graft and does not raise concerns about bone resorption or lack of incorporation. Consequently, we believed that MA has advantages, especially during the long-term follow-up period, when compared to ARR with allogenic bone graft. Indeed, several studies have reported favorable results when combining MA with allogenic bone graft. However, there is a lack of studies that have reported long-term follow-up results.

Following your comments, we have added the discussion about this issue more clearly, in Discussion section.

Text changes:

Porous metal augments offer several advantages over structural allografts such as easier surgical technique and no risk of disease transmission.[27, 43] In addition, porous metal augments are reliable for bone ingrowth and will not be resorbed.[43, 44] Consequently, MA adds stability onto the cage allowing for better survival. (page 8, line 235-239)

Recently, the introduction of porous tantalum MA has been a paradigm shift in revision THA. While serving as a void filler in multiple different shapes, it has an excellent innate properties that can promote both primary biologic fixation and supplementary fixation for adjunctive implants.[13] (page 2, line 58-61)

However, the bone grafts around ARR often result in failure due to bone resorption or lack of incorporation.[21, 24] Therefore, use of ARR alone has the risk of implant breakage and early loosening due to the lack of biologic incorporation. (page 2, line 54-57)

  1. The method shown or described by the authors is not novel. This technique is performed widely. The results are good.

Author’s response:

Thank you for your valuable comment. To be honest, when we initiated the study in 2015 and first submitted to another publisher in 2021, we mistakenly thought that this study was the first of its kind with mid to long-term follow-up period with detailed depiction of surgical procedures. Unfortunately, we realized more studies have been published since then, which undermines the originality of our study. Regardless, although small in number of cases, this study represents the longest follow-up period along with detailed depiction of surgical procedures in the current body of literature to the best of our knowledge. As you mentioned, while this technique may be widely used, there are only 3 studies available to date. Since more long-term follow-up results are needed for this new construct, we believe that our study still adds to a valuable collection of rare type of revision cases. 

Baecker, H., et al. (2020). "Tantalum Augments Combined with Antiprotrusio Cages for Massive Acetabular Defects in Revision Arthroplasty." Arthroplast Today 6(4): 704-709.

Garceau, S. P., et al. (2022). "Hip Arthroplasty With the Use of a Reconstruction Cage and Porous Metal Augment to Treat Massive Acetabular Bone Loss: A Midterm Follow-Up." J Arthroplasty 37(7s): S636-s641.

Mäkinen, T. J., et al. (2017). "Management of massive acetabular bone defects in revision arthroplasty of the hip using a reconstruction cage and porous metal augment." Bone Joint J 99-b(5): 607-613.

Text changes:

Three studies were used as references throughout the entire text.

  1. Overall well written and interesting. It is a good read for orthopaedic hip surgeons. 

Author’s response:

We appreciate your kind comments. We hope our point-by-point responses would meet your expectations. As you have advised, we have carefully revised the manuscript to improve its quality.

Reviewer 3 Report

Nice to see a non-TMARS cage being used successfully with TM augments.

I would like to see some statistical comment as to why there are no confidence intervals / limits on the Kaplan-Meier as it is difficult to interpret this graph without those.

I don't think figure 3b adds much, however a second case would be helpful as it is tempting to only show the best case.

Author Response

  1. Nice to see a non-TMARS cage being used successfully with TM augments.

Author’s response:

We sincerely appreciate your kind comments.

Text changes:

None

  1. I would like to see some statistical comment as to why there are no confidence intervals / limits on the Kaplan-Meier as it is difficult to interpret this graph without those.

Author’s response:

We appreciate your taking the time to review our paper and suggest constructive advices. Unfortunately, we removed the KM curve following Reviewer #1’s comment (“the sample size is small, Kaplan Meier curve seems meaningless”). We apologize that we could not add the additional CI / limits on KM curve due to the remove KM curve.

Text changes:

REMOVED figure 4.

  1. I don't think figure 3b adds much, however a second case would be helpful as it is tempting to only show the best case.

Author’s response:

Thank you for your valuable comment. We have added a new figure (Figure 4) with Case #3, which was considered a treatment failure due to the development of periprosthetic joint infection as postoperative complication.

Round 2

Reviewer 1 Report

Thank you for revising the manuscript. This paper seems to be well revised. I accept in present form.